# Vitamin B6 in Health and Disease

**DOI:** 10.3390/nu13093229

**Published:** 2021-09-17

**Authors:** Kamilla Stach, Wojciech Stach, Katarzyna Augoff

**Affiliations:** 1Department of Medical Biochemistry, Wroclaw Medical University, 50-368 Wroclaw, Poland; wojciech.stach@student.umed.wroc.pl; 2Department of Surgical Education, Wroclaw Medical University, 50-668 Wroclaw, Poland; katarzyna.augoff@umed.wroc.pl

**Keywords:** vitamin B6, pyridoxal 5′-phosphate (PLP), diabetes, COVID-19, cardiovascular disease, immune system, cancer

## Abstract

Vitamin B6 is a fascinating molecule involved in the vast majority of changes in the human body because it is a coenzyme involved in over 150 biochemical reactions. It is active in the metabolism of carbohydrates, lipids, amino acids, and nucleic acids, and participates in cellular signaling. It is an antioxidant and a compound with the ability to lower the advanced glycation end products (AGE) level. In this review, we briefly summarize its involvement in biochemical pathways and consider whether its deficiency may be associated with various diseases such as diabetes, heart disease, cancer, or the prognosis of COVID-19.

## 1. Introduction

Vitamin B6 is a very important molecule necessary for the health and proper functioning of the human body. It comprises a group of six water-soluble chemical compounds. These vitamins are pyridoxal (PL), pyridoxamine (PM), pyridoxine (PN), and their 5′-phosphates (Figure 1).

The active form, which is pyridoxal phosphate (PLP), serves as a cofactor for about 160 reactions in the body. So, the term “PLP” is used interchangeably with “vitamin B6” [1]. It is found in both eukaryotes and prokaryotes [1,2]. This compound is not produced by humans or other higher organisms, but yeasts and bacteria have the ability to produce it, although in a different way. By supplying them with food pyridoxal, pyridoxine, and pyridoxamine, humans can convert them, due to the presence of a specific pyridoxal kinase (PDXK), into active phosphates [3,4]. Vitamin B6 participates in the transformation of carbohydrates, lipids, amino acids, and nucleic acids. The best-known processes are glycogen breakdown, where it cooperates with glycogen phosphorylase; amino acid transformations, where it is a coenzyme in transamination and decarboxylation reactions; and reactions catalyzed by amino acid synthases or racemases. Descriptions of the mechanisms of these reactions can be found in any textbook on biochemistry. The key pathways for human health with the regulatory involvement of vitamin B6 include the metabolism of sphingosine phosphate [5], tryptophan catabolism [6], and the action of the transcription factor NF-κB. According to the latest research, it has the effect of reducing inflammation in the body by influencing the activity of inflammasomes and specifically its component, the NLRP3 sensory protein [7]. PLP controls processes such as blood pressure regulation (influence on the renin-angiotensin system) and blood clotting, ensuring endothelial integrity and platelet aggregation, all of which have an impact on human health and its disorders. The WHO-recommended daily intake of vitamin B6 for adults is 1.3–1.7 mg per day.

## 2. Some Metabolic Functions of PLP in Health

### 2.1. Role of PLP in Metabolism of S1P and Tryptophan

PLP is involved in the transformation of sphingosine-1-phosphate (S1P) formed in platelets but stored in red blood cells where it is protected against decomposition. S1P is a powerful inflammatory regulator, which plays a major role in the release of lymphocytes from the lymphoid organs. S1P is a product of the activity of sphingosine kinase and a regulator in the transformations of cells of the hematopoietic and nervous systems. It is called the conductor among particles, influencing their differentiation, migration, and adhesion, as well as their lifetime. The cells it acts upon have receptors for this compound on their surface. These are membrane-coupled receptors with G protein. The role of PLP is the degradation of S1P because it is a cofactor of S1P lyase (Figure 2).

When there is PLP deficiency, this lyase is inhibited, and when the lyase is deficient, lymphocytes are arrested in secondary lymphatic organs and in places where inflammatory processes take place [5,6]. This leads to lymphopenia and immunosuppression as well as to local exacerbation of inflammatory processes and to an increase in the secretion of pro-inflammatory cytokines [5]. Restoring the homeostatic function of the S1P system by replenishing PLP may allow an increase in immunity and maintaining an equilibrium that allows control of viral replication without uncontrolled expression of cytokines [6]. In the metabolism of tryptophan, PLP is a cofactor of enzymes in the kynurenine pathway. An example is kynureninase in its conversion to the ketoadipate pathway. This enzyme converts 3-hydroxykynurenine to 3-hydroxyanthranate (Figure 3). With PLP deficiency, xanthate is formed instead of 3-hydroxyanthranate. The level of this substance is an indicator of vitamin B6 deficiency in the body. In order to test its level, the tryptophan loading test is used. PLP deficiency, by impeding the catabolism of Trp to NAD^+^, reduces the formation of energy and resistance to stress [6,8,9]. Kynurenine metabolites accumulating in the case of PLP deficiency (including kynurenine, kynurenic acid, and 3-hydroxykynurenine) act via the aryl hydrocarbon receptor (AhR) and participate in inflammation. The AhR receptor is a transcription factor, which is involved in the regulation of cytochrome p450 levels but also in the regulation of immunity and cell differentiation. Kynurenine induces immunosuppression by reducing the activity of NK cells and the level of T lymphocytes, leading to an increase in the level of their apoptosis [9,10]. Kynurenic acid can stimulate the production of cytokines, in particular IL-1, IL-6, IL-10, and TNFα, and enhance the axis of the IL-6/JAK/STAT signaling pathway by acting on the principle of positive feedback with stimulation of IL-6. This, in turn, can stimulate inflammation [9,10]. PLP lowers the production of IL-1β, which is a potent proinflammatory cytokine, and the production of ROS by inhibiting the NLRP3 inflammasome. Vitamin B6 deficiency may result in increased production of IL-1β and tissue damage due to the presence of free radicals [7].

### 2.2. Role of PLP in Other Pathways

Another interesting role of PLP is the regulation of the enzyme serine hydroxymethyltransferase 2 (SHMT2), which is involved in the metabolism of folate. This transferase also regulates the production of type I interferons [11,12].

Moreover, PLP plays an important role in the homocysteine pathway as a cofactor of two cystathionine synthase enzymes: in the conversion of homocysteine to cystathionine and cystathionase, and in the synthesis of cysteine from cystathionine. The lack of this cofactor results in an increased level of homocysteine (HCys), which has serious health implications, such as the development of atherosclerosis [13,14].

Since vitamin B6 is present in so many reactions in the body, it is not surprising that its deficiency has significant clinical implications. This review tries to answer the question of whether a deficiency of vitamin B6 plays a role in developing diseases such as cancer, diabetes, immune disorders, inflammation, and other health problems, as well as in a more serious course of SARS-CoV-2 infection.

## 3. Vitamin B6 and Diseases

### 3.1. Vitamin B6 and Diabetes

High glucose concentration promotes the formation of advanced glycation products (AGEs) and oxidative stress, and thus damages some organs, mainly the heart, nerves, eyes, and kidneys. Nephropathy is the most common kidney damage in people with diabetes, characterized by albuminuria. Strict glycemic control, unfortunately, does not always allow avoidance of secondary diabetic complications and vice versa, as sometimes a patient who is ill for a long time is not susceptible to microvascular complications [15]. In order to prevent these complications, researchers started looking for other factors that predispose these complications—genetic, metabolic, or dietary. The supplementation of rats with vitamin B6 showed that albuminuria could be inhibited in this way [16]; however, the combination of vitamins B6 and B1 significantly reduced glycation of nuclear DNA in leukocytes. Nix et al. studied the effect of vitamin B6 levels in patients with type 2 diabetes, both with and without nephropathy. Their studies showed that type 2 diabetes is associated with a decreased level of vitamin and changes in its metabolism, especially in patients with initial nephropathy [17]. The PLP level was also lower in the group with or without retinopathy compared to healthy subjects [18]. Other metabolic forms of vitamin B6 have been rarely studied to date. Nix et al. found that pyridoxine and pyridoxal, like PLP, were present at a lower concentration, while the median levels of pyridoxine, PMP, and pyridoxic acid were much higher. This may indicate that type 2 diabetes is related to the variable activity of the enzymes involved in converting vitamin B6. Perhaps the reason is the decreased conversion to PMP in the reaction catalyzed by the flavin-dependent pyridoxine oxidase to PLP. The same oxidase also catalyzes the conversion of pyridoxine-phosphate to PLP. Most of the teams testing PLP levels found an inverse relationship between the level of PLP and the occurrence of diabetes. According to some teams, the more advanced the disease, the greater the dependence [17,18]. Studies by other authors have indicated an elevated level of vitamin B6 in the urine, which may, in turn, mean that its absorption is impaired [19]. Similar results were obtained when conducting experiments on rats [20,21]. It is not entirely clear whether lowered PLP levels contribute to the development of diabetes, or whether diabetes lowers PLP levels. Both of these hypotheses seem plausible. Leklem and Hollenback, in 1990, showed that glucose consumption by healthy people lowers PLP levels [22]. Okada proposed that in diabetes, this may be due to the increased rate of metabolism of PLP-mediated proteins in the body in a low-carbohydrate diet. They observed in diabetic rats on a low-PLP diet, a four-fold higher level of aspartate aminotransferase (AspAt) compared to the control group and a reduced level of glycogen phosphorylase [21]. As stated above, diabetes is associated with inflammation throughout the body [23] and perhaps the decreased level of PLP results from its involvement in pathways active during inflammation. Toyota et al., after conducting research on rats, stated that a reduced level of PLP may interfere with insulin secretion by the islets of the pancreas. Performing tests on the perfused pancreas, they observed impairment of insulin and glucagon secretion by the pancreas when there were deficiencies in pyridoxine [24]. PLP is also a cofactor of glutamate decarboxylase, in which GABA is formed. Antibodies to this enzyme are an important marker of diabetes. According to Rubí’s hypothesis [25], a decreased level of PLP may trigger autoimmunity processes that destroy the islets of the pancreas. A decrease in the level of PLP was also observed in gestational diabetes. In 2010, Oxenkrug noticed that the vitamin-B6-dependent conversion of tryptophan to serotonin is disturbed in pregnant women. Although studies [26] suggested that vitamin B6 deficiency increases the risk of glucose intolerance in pregnancy and supplementation improves the regulation of these pathways, only Fields conducted experiments in which they checked whether vitamin B6 deficiency in pregnancy disturbs Trp metabolism and activation of the serotonin HTR2B receptor in the pancreas. This, in turn, would have an impact on the proliferation of beta cells and thus reduced insulin secretion, which causes gestational diabetes. The studies were carried out on pregnant mice that were deficient in dietary vitamin B6, after which their glucose levels were measured and a glucose load test was performed. Control groups were mice before pregnancy and three weeks after birth. It was found that in the control groups, vitamin B6 deficiency did not significantly affect either the fasting glucose level or the sugar curve. Glucose intolerance with insulin resistance was present in PLP-deficient mice, but insulin levels remained unchanged. Probably this phenomenon is driven by other mechanisms and it requires additional research [27].

### 3.2. Vitamin B6, Immunity and Inflammation

Calder et al. [28], writing about the optimal nutritional status in the context of immunity, mention vitamin B6 as a compound that participates in the functioning of the entire immune system. On par with supporting immunity are vitamins A, B12, C, D, and E; folic acid; and trace elements, including zinc, iron, selenium, magnesium and copper. All these compounds support both innate and acquired immunity. According to these authors, the majority of the population has numerous deficiencies of these substances and their supplementation is recommended. The role of vitamin B6 in alleviating the symptoms of COVID-19 infection and complications such as diabetes, hypertension and heart disease after COVID-19, was reported by Kumrungsee [29]. Possible mitigation mechanisms may include inhibition of inflammation (stopping the cytokine storm) and oxidative stress, regulation of Ca^2+^ levels, increasing carnosine levels (as a cardio protector), and improving immune system function. Vitamin B6 plays a key role in the production of T lymphocytes and interleukins. Its deficiency leads to a decrease in immunity, including the formation of serum antibodies, decreased production of IL-2, and increased IL-4 [30]. There is an inverse relationship between vitamin B6 and IL-6 and TNF-α levels in conditions of chronic inflammation. Patients suffering from COVID-19, who additionally have severe inflammation, may be deficient in it. Similarly, in elderly patients, as well as those with type 2 diabetes and cardiovascular disease, lower levels of vitamin B6 are noted [29,30]. It is extremely interesting that PLP also influences the formation of the intestinal microbiota. The composition of the microbiota, on the other hand, affects human immunity. As mammals do not synthesize B6, they are supplied with food and some intestinal bacteria that pose pathways for its synthesis (e.g., *Bacteroides fragilis, Prevotella copri, Bifidobacterium longum, Collinsella aerofaciens,* or *Helicobacter pylori*). Some of the microbiota’s organisms also need to obtain vitamins from food consumed by their host or from other gut bacteria. The bacteria that do not synthesize B6 include, among others, *Veillonella, Ruminococcus, Faecalibacterium,* and some *Lactobacillus* spp., and do not have the vitamin B6 biosynthetic pathway [31,32]. Sakakeeny tested 2229 patients for PLP levels and found that the lowest levels of this vitamin were in people who had chronic inflammation. Conversely, people with high levels of this vitamin had low levels of inflammation [33]. Some researchers also show that the absence of vitamin B6 in the body can have serious consequences for the immune system. Vitamin B6 deficiency disrupts the Th1–Th2 balance toward an excessive Th2 response, which may result in allergies [7,30,34]. Gombart et al. and many others give examples in their publications of the role of vitamin B6 in infections, immune response, enhancing immunity in the intestines, and enhancing the cytotoxic activity of NK [4,33,35].

### 3.3. Vitamin B6 and Cancer

Chronic inflammation in the body can lead to neoplastic processes. In recent years, there has been a lot of research on the effects of vitamin B6 on inflammation and chronic inflammation, including cancer. Mikkelsen [36] investigated the effect of B vitamins on the modulation of the immune response. They found that vitamins B2 (riboflavin), B6 (pyridoxine), and B9 (folic acid) exerted antitumor activity on promonocytic lymphoma cell lines. These results could form the basis for future research into the use of vitamin B supplementation to reduce cancer cell growth in vivo. Potential mechanisms underlying the anti-proliferative and anti-migratory effects of vitamins may include angiogenesis, altered cytokine secretion, altered PD-L1 expression (as a programmed cell death ligand), oxidative stress, and nitric oxide synthesis. In fact, vitamins B2, B6, and B9 all increased the secretion of IL-8 and IL-10 by cells of the U937 monoblastic leukemia cell line compared to control groups [36]. PLP is a cofactor for critical enzymes in the methyl metabolism pathways. Vitamin B6 plays a key role in the kynurenine pathway as a cofactor of the enzyme kynureninase. This pathway produces anti-inflammatory molecules such as kynurenine [5]. Research shows that an optimized level of kynureninase with its cofactor can inhibit progression of cancer in vivo [37]. An end-metabolite of kynurenic axis, 8-hydroxyquialdic acid has anti-proliferative and anti-migratory effects on colorectal cancer cells. Kynureninase activity is regulated by the level of PLP [37]. The influence of vitamin B6 on the development of cancer is also a popular topic for meta-analysis. Wei and Mao [38] worked with data of vitamin B6 intake and high plasma PLP levels as protectors against pancreatic tumor. A review by Mocellin [39] states that there is an undoubtedly strong relationship between vitamin B6 intake and the appearance of cancer, regardless of where it occurs. Their work shows clearly that there is an inverse linear relationship between cancer risk and both vitamin B6 dietary intake and PLP levels. The strongest connection occurs in gastrointestinal cancer. Both works ask the same question: can this vitamin directly prevent cancer or is it just a marker in healthy humans’ plasma and an indicator of vitamin-rich food? The conclusions are not yet quite clear, and both hypotheses require further research.

### 3.4. Vitamin B6 and Cardiovascular Diseases and Pneumonia

One of the functions of vitamin B6 that is widely discussed is its role in CVDs and in lowering of blood pressure. Low plasma levels of PLP in humans are also associated with high-risk atherosclerosis, stroke, and thrombosis [7,39,40]. Severe vitamin B6 deficiency is rare, while suboptimal levels or slight deficiency are more common. Deluyker [41], in studies conducted on rats, investigated whether PM, which inhibits the formation of AGEs, decreases collagen level and improves cardiac function. Jeon and Park [42] conducted a study of the effects of B6 on a large group of Koreans. They observed that increased doses of vitamin B6 ingested lowered the risk of developing CVDs in men, but that no such relationship occurred in women [42]. Page [43] conducted studies on the influence of concentration of PLP on post-menopausal women. The conclusion was that the concentration in plasma is inversely related to the risk of myocardial infarction. Hellman and Mooney investigated the mechanism of the cardio-protective action of vitamin B6 [1]. The reason is still unclear but may be because folates and cobalamin lower the level of HCys. This amino acid is transferred into cysteine by cystathionine synthase, as mentioned earlier, and PLP is a cofactor of this enzyme. It is known that a high level of HCys is a risk factor for atherosclerosis and high blood pressure [13]. Chronic inflammation is also a key mechanism underlying atherosclerosis and its progression. Plasma levels of PLP are inversely correlated with systemic markers of inflammation, e.g., CRP [40]. Vitamin B6 supplementation decreased IL-6 levels and increased total lymphocytes in patients with chronic disease [44]. Vitamin B6 may also regulate Ca^2+^ influx into cells via voltage-gated ATP-dependent purinergic receptors, suggesting its role in the regulation of hypertension and cardiac dysfunction [45]. In recent years, a new function of vitamin B6 in protecting the heart has also been identified: imidazole dipeptides. PLP regulates the homeostasis of, for example, carnosine, homocarnosine, and anserine, which are cardio protectors with antioxidant and anti-inflammatory properties. Most likely, enzymes involved in the synthesis of β-alanine, i.e., a precursor compound for carnosine, depend on pyridoxal phosphate [46,47]. The latest literature review by Kumurgsee [48] not only confirms the contribution of vitamin B6 to cardio protection through the action of histamine, GABA, and imidazole dipeptides, but also through inhibition of the P2 × 7R-NLRP3 inflammasome. Moreover, the modulation of anserine, carnosine, histamine, GABA, and the P2X7R–NLRP3 inflammasome may be involved in reducing inflammation and oxidative stress.

Shan et al. tested whether vitamin B6 supplementation could prevent pneumonia. The team investigated the anti-inflammatory effect of vitamin B6 in mice with acute pneumonia induced by administering LPS. Inflammation was measured by the levels of IL-1β, IL-6 and TNF-α. Researchers have shown that the vitamin can stimulate the phosphorylation of AMPK at the Thr172 residue, thereby stimulating its activity. This kinase, in turn, is an inhibitor of inflammatory processes in the cell. Thus, the researchers postulated vitamin B6 supplementation in the treatment of acute pneumonia. Knockout mice for the DOK3 gene (i.e., knockout mice for AMPK) were tested and it was observed that the anti-inflammatory effects of vitamin B6 in LPS-treated macrophages were abolished. However, they were enhanced in macrophages over-expressing DOK3. Hence, it was postulated that vitamin B6, by activating the AMPK-DOK3 pathways, could become a drug preventing pneumonia in people [49].

### 3.5. Vitamin B6 and COVID-19

After entering the body, SARS-CoV2 virus binds to the angiotensin converting enzyme (ACE) 2 and is highly expressed, especially in the alveoli, respiratory epithelia and endothelial cells of the heart and vessels [50]. Another feature of COVID-19 patients is the cytokine storm resulting from abnormal and excessive patient immune responses. This explains such common complications in people with hypertension and CVDs. Serum tests of patients infected with COVID-19 showed elevated levels of inflammatory markers such as CRP, IL-6 and lymphopenia. These are typical parameters for the severity of the disease. Chronic inflammatory and immune-compromised diseases, such as diabetes, increase susceptibility to viral infections [51]. Research suggests that low dietary intake of vitamin B6 is associated with a high risk of death from CVD, and supplementation with vitamin B6 reduces this risk [39]. According to Lengyel et al. [52], the intake of many nutrients, including PLP, is too low. Low PLP levels were also noted in patients with type 2 diabetes and in people with CVDs [17,53], that is, in groups with a higher risk of worse effects of COVID-19. Dysregulation of the immune response and an increased risk of coagulopathy have also been reported in COVID-19 patients [54]. Since vitamin B6 supplementation causes a reduction in blood pressure in hypertensive patients, Kumrungsee et al. postulated that it may also alleviate the symptoms of COVID-19, leading to a reduction of complications by the above action [29]. During COVID-19 infection, when the consumption of PLP is increased, the more we understand about the deficiencies of this vitamin and, consequently, about the dysregulation of immune responses. Australian researchers reported that vitamin B6 (as well as B2 and B9) increases levels of IL-10, a potent anti-inflammatory and immunosuppressive cytokine that can inactivate macrophages and monocytes and inhibit antigen-presenting cells and T cells. Patients with COVID-19 often respond to the virus by enhancing the excessive response of T lymphocytes and secretion of pro-inflammatory cytokines. It is possible that PLP may contribute to the suppression of the cytokine storm and inflammation from which some COVID-19 patients suffer [54]. Camara [55] reported on the European Commission’s EU Register of Nutrition and Health Claims Made on Foods authorized data of vitamin B6 related to the function of the immune system. The amount of the vitamin that helps support a human’s immune system is 0.21 mg/100 g (foods) and 0.105 mg/100 mL (beverages).

### 3.6. Some Other Aspects of Vitamin B6

Schorg [56] compared the status of vitamin B6 in vegetarians, and flexitarians, and pescatarians to people who eat meat. Although the vitamin from plants has lower bioavailability, vegetarians do not have a risk of deficiency. Berkins [57] tried to understand whether an association exists between depression and deficiency of vitamin B6 (testing B12 and folate as well). While they observed some associations between these vitamins and brain volume, these differences were not significant, although depression was more prevalent in vegetarians (48.7%) compared to non-vegetarians (32.6%). The authors postulate that vegetarians with depression may benefit from supplementing these vitamins for brain health, but more research is needed. Nemazannikowa [58] investigated whether there is an association between vitamin B and sclerosis multiplex. Although deficiency of pyridoxine is not common, people with such deficiency suffer from depression, confusion and irritability, impaired immune system, and inflammations [59]. They conclude that impaired immunology response between patients with sclerosis multiplex is correlated with the level of pyridoxine and delayed hypersensitivity reactions along with decreased production of antibodies. More research is needed in this field.

## 4. Conclusions

A brief review of the literature related to the function of vitamin B6 in metabolic changes, and thus its participation in maintaining health, proves that it is a molecule necessary for the proper functioning of the entire body and its role cannot be overestimated. Figure 4 shows some diseases in which vitamin B6 deficiency was observed. We consider that researchers are just a step away from finding the perfect combinations of vitamins, small molecule supplements, or micronutrients that are most beneficial for specific diseases.

## Figures and Tables

**Figure 1 nutrients-13-03229-f001:**
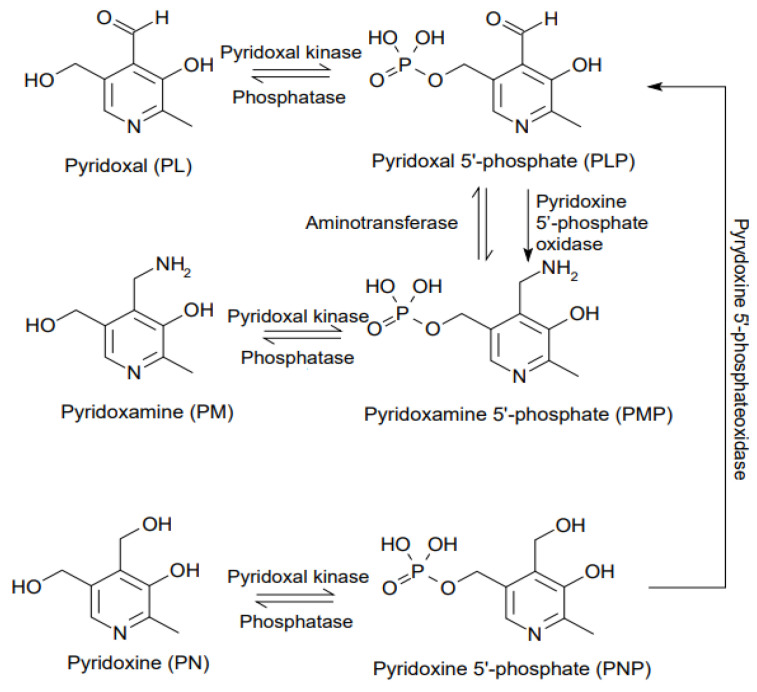
Forms of vitamin B6.

**Figure 2 nutrients-13-03229-f002:**
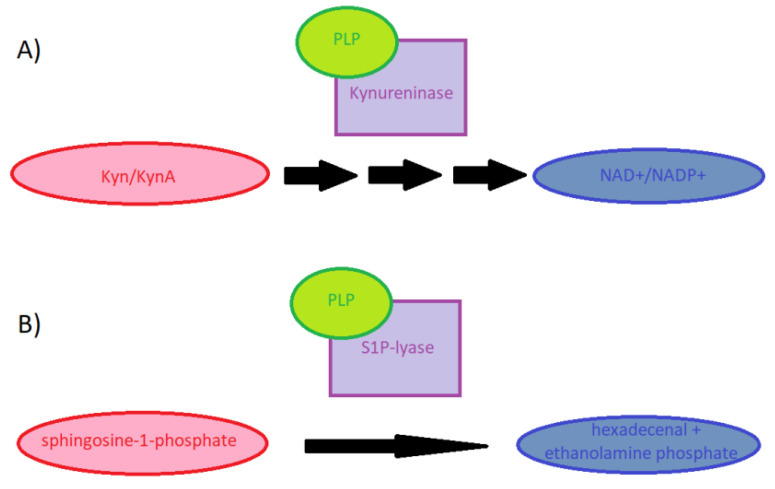
The role of PLP in metabolism of (**A**) tryptophan and (**B**) sphingosine-1-phosphate (S1P). Abbreviations: Kyn: kynurenine, KynA: kynurenic acid, NAD^+^: nicotinamide adenine dinucleotide, NADP^+^: nicotinamide adenine dinucleotide phosphate.

**Figure 3 nutrients-13-03229-f003:**
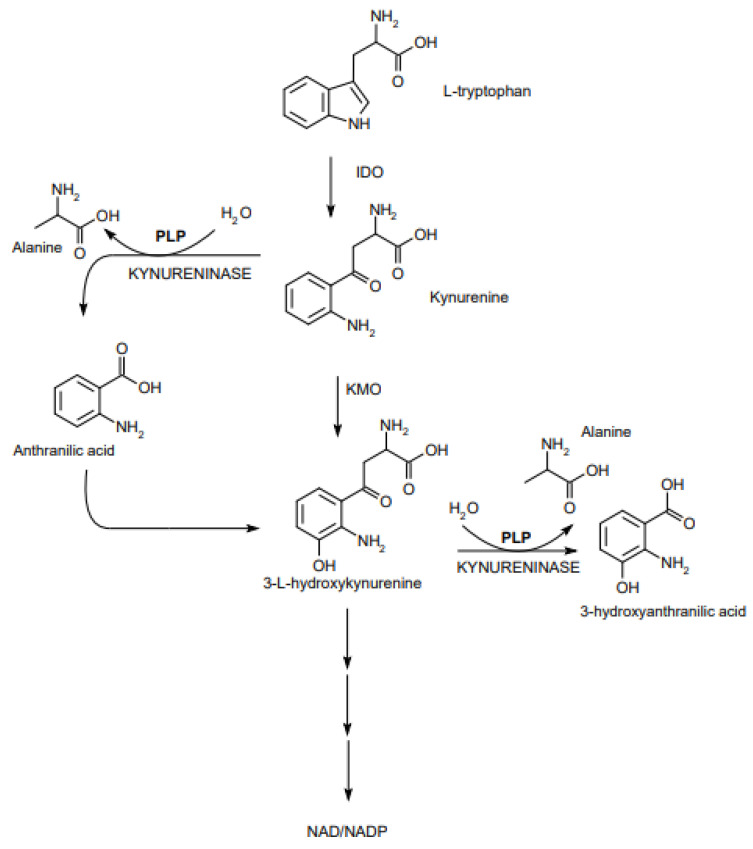
A role of PLP in the kynurenine pathway.

**Figure 4 nutrients-13-03229-f004:**
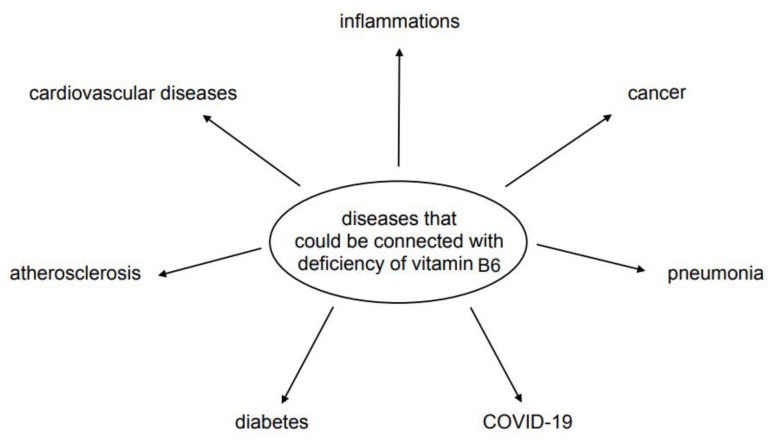
Possible diseases related to vitamin B6 deficiency.

## Data Availability

Not applicable.

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
