# Peer review of "Vitamin B6 in Health and Disease"

_nutrients, 2021, doi:10.3390/nu13093229_

Round 1

Reviewer 1 Report

The review aims to describe the role of vitamin B6 in human health. After a brief introduction on the possible vitamin isoforms and the biochemical role of the molecule, the authors propose some literature data with particular regard to the role of vitamin b6 on metabolic pathologies and the possible role in covid-19.

The article is not very smooth, perhaps due to linguistic problems. Animal data are often mixed with human data and frequently forcing. Many parts need references that are missing or perhaps are proposed late concerning the concepts expressed.

Here are some tips for the authors to implement the manuscript:

- In the introduction, it would be useful to propose a figure with the structures of the isoforms and, later in the text, figures that summarize the metabolic pathways and cellular interactions of the molecule. More generally, it is always better for a review paper to be accompanied by at least one figure and a table, especially when the extension is very small.

- Speaking of vitamin B6, deficiency stages, nutritional sources and groups of the population at risk of deficiency should be treated with more attention.

- Why is paragraph 2 called “Results”? What results are the authors referring to? It is advisable to implement the manuscript by entering the selection method used for the sources (search engines, terms used, filters, etc.).

The paragraphs' organisation is not clear: "2.results", "3. Vitamin B6 and diseases ". Furthermore, paragraph 2 has only one subparagraph.

Long periods have no bibliographic references (lines 46-56, lines 71-84, 97-100, lines 162-170, and 200-203). It is recommended to implement these parts with adequate references.

- The paragraph regarding covid-19 is very forced. It is recommended to implement it with significant results. Alternatively, it would be more correct to remove it.

- It is advisable to have the manuscript corrected by a native speaker and to reread the text carefully as many sentences would need to be rearranged

Minor aspects:

- On lines 25 and 27 the term "man" should be replaced with "humans"

- Reword the sentence in line 54 "when there is too little plp"

- The period on lines 71-75 is duplicated

- On line 93 the dot with a colon should be replaced

- On line 105 there is a non-English term

- There are extra spaces on lines 126, 150, 181,237

- On line 128 the reference should be formatted like the others (number in square brackets).

- On line 136 a typo in the word "befoere"

- Line 186 is missing a parenthesis

- On line 276 the term "Postulate" should is in lowercase

- On line 294 only "The authors declare no conflict of interest." is needed

Author Response

Dear Professor,

Thank you very much for your critical look at this article on „Vitamin B6 in Health and Disease”. I tried to follow all the comments for which I am very grateful. I did everything to improve the work so that the English language was scientific and understandable. Acording your sugestio nit was readed by native speaker. I have put papers on animals only where necessary due to the possibilities they offer and the continuation of my research. I tried to include literature references wherever indicated.

In line with your suggestion, I have attached the diagrams to the publication to make the work clearer and easier to read.

The fragments on deficiencies or groups at risk of deficiency have been corrected and reference was made only to the literature.

Thank you very much for your attention to the layout of paragraphs, it was an oversight on my part. I hope that this is fine now.

As for the COVID, it remained in the paper because, as suggested by the second reviewer, I corrected it and completed it with  the latest literature.

All minor aspects were fixed as suggested. Thank you very much for these comments. If I can somehow improve this publication, please kindly provide other comments. I will do it immediately.

Sincerely Yours,

Kamilla Stach

Reviewer 2 Report

This work by Stach et al brings a number of unique perspectives of Vit B6’s beneficial effects in health and disease, however the overall major improvement in writing is needed.

1.Title: All the vitamins essentially are pro-health. Thus, it is non-essential to state in the title that Vitamin B6 is prohealth.

  1. Please restructure the sentence in line 19. The objective of a review is to present, summarize and synthesize the facts with supporting references not to convince readers, per se.
  2. Authors should avoid using first-person narratives. Ex: Line 23
  3. Reference needed in support of line 24
  4. Sentece restructure Line 32-33
  5. As this is a review so avoid using subtitle Results
  6. Abbreviation correction S1P, Line 47

8.In support of lines 55-56 provide reference along with research article reference.

  1. Line 79-82 Reference needed
  2. Line 78, abbreviation correction ( AhR)
  3. A schematic of how the PLP-S1P-Kynurenine axis is interconnected would help readers to understand better
  4. Line 113, Instead of Nix studied mention as Nix et al, likewise in other lines such as 118,131,133,138,146,150, 163, etc.

13.Spelling correction line 136,156

  1. Sentence restructuring needed in line 167: avoid using words such as ‘intended to support’ instead suggested can be used.

15.Subtitle 3.2 : more explanation and detailed information needed on the role of Vitamin B6 in specific adaptive and innate immune cells. As in line 194 mentioned Vit.B6 deficiency disrupts Th1-Th2 balance, how?  Specific interleukin, chemokine receptor expression in adaptive and innate immune cells related to Vit. B6 bioavailability and how it affects immune cell recruitment and inflammatory responses?

  1. Line 203,205,209 reference needed.
  2. Further explanation is needed, B6 is a cofactor for which enzymes, Examples of how Vit B6 regulates relevant enzymes affecting methyl metabolism in specific cancer.
  3. References needed in Line 220, 223, 226.
  4. Reference needed line 237
  5. Line 251: Ca+2 influx instead of Ca influx
  6. More detailed information is needed to be incorporated in the role of Vit B6 in lung disease pathologies and cardiac pathologies since the subtitle is focused on cardiovascular systems. What is the role of Vit B6 in inflammation-associated angiogenesis or lymphangiogenesis? More specifically whether VitB6 has a favorable effect on lymphatic vascular aging? In addition, what is the effect of Vit B6 in lymphangiogenesis in lung diseases?
  7. Whether Vit B6 has a beneficial role in the progression of long covid pathophysiology?

All the references are not arranged in a uniform manner, for example between references 3 and 4.

In addition, it is recommended to avoid the use of abbreviations in abstracts. For example, AGE, In abstract line 11.

Author Response

Dear Professor,

Thank you very much for your critical look at this article on changed tittle „Vitamin B6 in Health and Disease”. I tried to follow all the comments for which I am very grateful. I did everything to improve the work.

  1. I rescruturred the sentence in line 19.
  2. Of course, first-person use of the phrase is unacceptable and this has been corrected.
  3. Reference was added.
  4. The sentences in 32-33 lines were changed.
  5. Thank you very much for your attention to the layout of paragraphs, it was an oversight on my part. I hope that this is fine now.

7, 10 Abbreviations  are corrected now.

8, 9, 12, 16, 18, 19 - I tried to complete all the footnotes and I hope that now there is nothing missing.

  1. the schemat how the PLP acts on kynurenine and PLP is now included.
  2. All the literaturÄ™ changes were made acording to your sugestions.

13, 20. Spelling mistakes were corected.

  1. Senstnce in line 167 were rescructurised.

15 and 17. Subtitles 3.2  and 3.3  this subsection has been revised and corrected as suggested

21 and 22. As you  suggested  I corrected it and completed it the paragraph concerning CVD and COVID-19 with  the latest literature and  to show if  does vitamin b6 has bebefitial role in pathophysiology of COVID.

All aspects were fixed as suggested.

Thank you very much for these critical comments and advices .  If I could still improve the quality of the publication in some way, please give me some tips. I will do it immediately.

Sincerely Yours,

Kamilla Stach

Round 2

Reviewer 1 Report

The authors improved the manuscript with in-depth information and a smoother structure

Author Response

Dear Professor, thank you very much for your time and valuable comments. I tried very hard to adapt to all the comments. Currently, I have also corrected the linguistic errors and completed the literature data.
I kindly ask you to accept the work.

Kind regards,
K. Stach

Reviewer 2 Report

Thank you for addressing the concerns. Minor revision needed for spelling check and for the reference in the text use author name et al format.

Author Response

(The authors gave the same response as above.)
